# Uncertainty-based Graph Convolutional Networks for Organ Segmentation Refinement

**Roger D. Soberanis-Mukul**[1]                                        ROGER.SOBERANIS@TUM.DE

**Nassir Navab**[1,2]                                              NASSIR.NAVAB@TUM.DE

**Shadi Albarqouni**[1,3]                                        SALBARQOUNI@ETHZ.CH

[1] *Computer Aided Medical Procedures, Technische Universität München, Germany*

[2] *Computer Aided Medical Procedures, Johns Hopkins University, Baltimore, USA*

[3] *Computer Vision Laboratory, ETH Zurich, Switzerland*

**Editors:** Under Review for MIDL 2020

## Abstract

Organ segmentation in CT volumes is an important pre-processing step in many computer assisted intervention and diagnosis methods. In recent years, convolutional neural networks have dominated the state of the art in this task. However, since this problem presents a challenging environment due to high variability in the organ's shape and similarity between tissues, the generation of false negative and false positive regions in the output segmentation is a common issue. Recent works have shown that the uncertainty analysis of the model can provide us with useful information about potential errors in the segmentation. In this context, we proposed a segmentation refinement method based on uncertainty analysis and graph convolutional networks. We employ the uncertainty levels of the convolutional network in a particular input volume to formulate a semi-supervised graph learning problem that is solved by training a graph convolutional network. To test our method we refine the initial output of a 2D U-Net. We validate our framework with the NIH pancreas dataset and the spleen dataset of the medical segmentation decathlon. We show that our method outperfroms the state-of-the art CRF refinement method by improving the dice score by 1% for the pancreas and 2% for spleen, with respect to the original U-Net's prediction. Finally, we discuss the results and current limitations of the model for future work in this research direction. For reproducibility purposes, we make our code publicly available[1].

**Keywords:** Organ segmentation refinement, Uncertainty Quantification, Graph Convolutional Networks, Semi-Supervised Learning

## 1. Introduction

Segmentation of anatomical structures is an important step in many computer-aided procedures, like medical image navigation and detection algorithms. Many of these methods rely on manually segmented inputs performed by clinical experts. However, this is a time-consuming task due to the large amount of information (generally volumes) that is generated. Organ segmentation in CT or MRI slices has been a topic of research for many years. Recently, with the growth of deep learning models, many architectures have been proposed for dealing with this problem. One of these challenges is related to the similarity between organs and background yielding to misclassifications, mainly in boundary regions

---

1. https://github.com/rodsom22/gcn_refinement

of the organs resulting in many false positives (FP) and false negatives (FN) regions. Such a problem hinders the model integration into clinical practice, where higher precision is required. One way to improve the model performance is by introducing a post-processing refinement step in the pipeline.

Recent segmentation models for medical structures are based on convolutional neural networks (CNN). These models can be composed of aggregations of multiple 2D CNN (Zhou et al., 2017a; Roth et al., 2018b) or by 3D CNN (Zhu et al., 2018; Roth et al., 2018a). Similarly, models that incorporate shape and geometric priors have been recently proposed (Zhou et al., 2019; Yao et al., 2019; Degel et al., 2018). Refinement strategies are typically introduced at the end of the process to improve the model's prediction. This can also be used as an intermediate processing step, where more complex strategies can use the refined results to improve the segmentation. For example, in (Wang et al., 2018), a set of scribbles is generated by defining a conditional random field (CRF) problem that is solved with Graph Cut methods. These results can be combined with user-defined scribbles to perform an image specific fine-tune of a CNN segmentor. In another context, given the limited availability of labeled medical data, semi-supervised learning methods define strategies to include the (most commonly) available unlabeled medical data. Such strategies include the generation of pseudo-labels for unlabeled data. Here, refinement methods, like densely connected CRF (Bai et al., 2017) are included in the semi-supervised steps, to refine the pseudo-labels. Uncertainty has also proved to be useful as an attention mechanism in semi-supervised learning (Xia et al., 2018) and recent works in computer vision have started to explore the capabilities of uncertainty for finding potential misclassified regions for segmentation refinement purposes (Dias and Medeiros, 2019). In the medical context, uncertainty has been employed as a measure of quality for the segmented output (Roy et al., 2018), and its ability to reflect incorrect predictions has been recently studied (Nair et al., 2018). A recent work, presented by (Yu et al., 2019), uses the uncertainty of a teacher model to select the pseudo-labels to train a student model.

Even though dense graph representations of three-dimensional data have been applied for refinement (Kamnitsas et al., 2017), the use of recent graph convolutional networks (GCN) with sparse graphs representations of 3-D data has not been fully investigated. In this paper, we propose a two-step approach for the refinement of volumetric segmentation coming from a CNN. First, we perform an uncertainty analysis by applying Monte Carlo dropout (MCDO) (Kendall and Gal, 2017) to the network to obtain the model's uncertainty. This is used to divide the CNN output in high confidence background, high confidence foreground and low confidence points (FP and FN candidates). The uncertainty is also used to define a 3-D shape-adapted region of interest (ROI) around the organ. With this information, we define a semi-labeled graph inside the ROI. We use this graph to train a GCN in a semi-supervised way using the high confidence predictions as labeled training nodes for the GCN. The refined segmentation is obtained by evaluating the full graph in the trained GCN.

*Contributions*: To our best knowledge, this is the first time a semi-supervised GCN learning strategy is employed in the medical image segmentation task, specifically, for single organ segmentation. Also, this work presents one of the first cases of using GCN and uncertainty analysis for segmentation refinement. We provide a framework in which a per voxel segmentation refinement task is mapped into a semi-supervised graph classification

problem. Thanks to the Monte-Carlo dropout estimation, voxels with high confidence are treated as labelled nodes in the graph, while the rest are treated as unlabelled nodes, and the main objective is to learn a GCN model to classify the unlabeled ones. Our framework operates on top of a CNN in inference time and it does not need to retrain the network.

## 2. Methods

*Overview*: Consider an input volume $V$ with $V(x)$ the intensity value at the voxel position $x \in \mathbb{R}^3$; consider also, a trained CNN $g(V(x); \theta)$ with parameters $\theta$; and a segmented volume $Y(x) = g(V(x); \theta)$ with $Y(x) \in \{0, 1\}$. Our objective, is to refine the segmentation $Y$ using a graph convolutional neural network (GCN) trained on a graph representation of the input data. Our framework operates as a post-processing step (one volume at a time) and assumes that no information about the real segmentation (ground truth) is available.

We first look for a binary volume $U_b$ used to highlight the potential false positives and false negatives elements of $Y$. The second step uses $U_b$, together with information coming from $Y$, $g$, and $V$, to refine the segmentation $Y$. We use uncertainty analysis to define $U_b$. For the second step, we solve the refinement problem using a semi-supervised GCN trained on a graph representation of our input volume.

### 2.1. Uncertainty Analysis: Finding Incorrect Elements

In our framework, incorrect elements are estimated considering the confidence of $g$. We employ MCDO approximation (Kendall and Gal, 2017) to evaluate the uncertainty of the CNN. This strategy can be applied to any model trained with dropout layers, without modifying or retraining the model. This attribute makes it ideal for a post-processing refinement algorithm. MCDO uses the dropout layers of the network in inference time, and perform $T$ stochastic passes on the network to approximate the output of a Bayesian neural network. Following this method, we get the model's expectation

$$\mathbb{E}(x) \approx \frac{1}{T} \sum_{t=1}^{T} g(V(x), \theta_t), \tag{1}$$

with $\theta_t$ the model parameters after applying dropout in the pass $t$. The model uncertainty $\mathbb{U}$ is given by the entropy, computed as

$$\mathbb{U}(x) = H(x) = -\sum_{c=1}^{M} P(x)^c \log P(x)^c, \tag{2}$$

with $P(x)^c$ the probability of the voxel $x$ to belong to class $c$, and $M$ is the number of classes ($M = 2$ in our binary segmentation scenario). We use $\mathbb{E}$ as an approximation of the probability volume $P$ for computing the entropy. Finally, we define the potential incorrect elements by applying a binary threshold on the entropy volume

$$U_b(x) = \mathbb{U}(x) > \tau, \tag{3}$$

where the uncertainty threshold $\tau$ controls the entropy necessary to consider a voxel $x \in Y$ as uncertain.

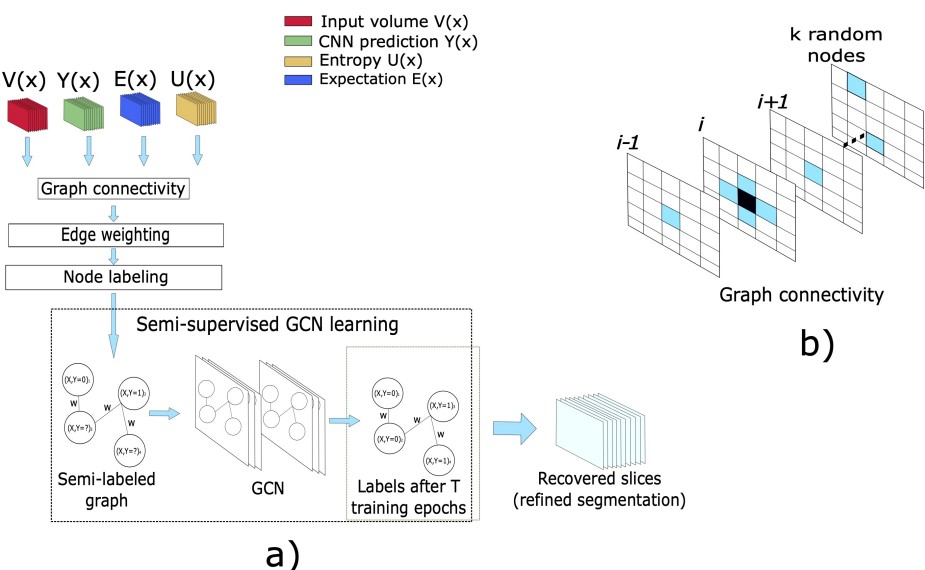

Figure 1: a) The GCN refinement strategy. We construct a semi-labeled graph representation based on the uncertainty analysis of the CNN. Then, a GCN is trained to refine the segmentation. b) Connectivity. The black square is connected to six perpendicular neighbors and with $k = 16$ random voxels

## 2.2. Graph Learning for Segmentation Refinement

At this point, we have a binary mask $U_b$ indicating voxels with high uncertainty. The uncertainty analysis only tells us that the model is not confident about its predictions. Some of the elements indicated by $U_b$ could be indeed correct and its value should not be changed. However, we can use a learning model that trains on high confidence voxels to reclassify (refine) the output of the CNN $g$. Using the information from the uncertainty analysis, we can define a partially-labeled graph, where the voxels are mapped to nodes, and neighborhood relationship to edges. In this way, we formulate the refinement problem as a semi-supervised graph learning problem. We address this mapped problem by training a GCN on the high confidence voxels using the methods presented in (Kipf and Welling, 2017). The rest of this section describes the formulation of our partially-labeled graph.

### 2.2.1. PARTIALLY-LABELED NODES

Given a graph $\mathcal{G}$ representing our 3D volumetric data, at the inference tine, we aim to obtain a refined segmentation $Y^*$ as the results of our GCN model $\Gamma$,

$$Y^* = \Gamma(\mathcal{G}(S); \phi), \tag{4}$$

where the graph $\mathcal{G}$ is constructed from the set of volumes $S = \{\mathbb{E}, \mathbb{U}, V, Y\}$ (see section 2.1 and Fig. 1) and $\phi$ represents the GCN's parameters.

Since most of the voxels in the volume are irrelevant for the refinement process and given that graphs are not restricted to the rectangular structured representation of data,

we define an ROI tailored to our target anatomy. We define our working region as $\text{ROI}(x) = \text{dilation}(U_b(x)) \cup \mathbb{E}_b(x)$ with $\mathbb{E}_b$ the expectation binarized by a threshold of 0.5. Since the entropy is usually high in boundary regions, including the dilated $U_b$ ensures that the ROI is bigger enough to contain the organ. Also, this allows us to include high confidence background predictions $(Y = 0)$ for training the GCN. Including the expectation in the ROI give us high confidence foreground predictions for training the model. This ROI reduces the number of nodes of the graph and, in consequence, the memory requirements. The voxels $x \in \text{ROI}$ define the nodes for $\mathcal{G}$. Each node is represented by a feature vector containing intensity $V(x)$, expectation $\mathbb{E}(x)$, and entropy $\mathbb{U}(x)$. Finally, we labeled each node in the graph according to its uncertainty level using the next rule:

$$l(x) = \begin{cases} Y(x) & \text{if } U_b(x) = 0 \\ \text{unlabeled} & \text{if } U_b(x) = 1 \end{cases} \tag{5}$$

### 2.2.2. Edges and Weighting

The most straightforward connectivity option is to consider the connectivity with adjacent voxels (6 or 26 adjacent voxels). However, this simple nearest neighborhood scheme may not be adequate in our problem for two reasons; First, with this scheme, every single voxel is connected with its local neighborhood but lacks global information. Second, voxels with high uncertainty tend to shape contiguous clusters. With a simple nearest neighborhood scheme, voxels inside these clusters will be only connected to their adjacent neighbors, i.e. uncertain voxels, with almost no connection with voxels with high confidence. Voxels living in the boundary of these clusters are the only ones who are connected to voxels with high confidence. Hence, the propagation of information from confidence to the uncertain regions will be somehow limited. A fully-connected graph can take advantage of the relationships between certain and uncertain regions in training and inference time but at a cost of prohibitive memory requirements. In our work, we evaluate an intermediate solution. For a particular node (or voxel) $x$, we create connections with its six perpendicular immediate neighbors in the volume coordinate system. Additionally, we randomly select $k = 16$ nodes in the graph and create a connection between these random elements and $x$. This defines a sparse representation that considers connections between labeled and unlabeled elements.

To define the weights for the edges, we use a function based on Gaussian kernels considering the intensity $V(x)$ and the 3-D position $x \in \mathbb{R}^3$ associated with the node:

$$w(x_i, x_j) = \lambda \text{div}(x_i, x_j) + \exp\left(-\frac{||V(x) - V(x_j)||^2}{2\sigma_1}\right) + \exp\left(-\frac{||x_i - x_j||^2}{2\sigma_2}\right) \tag{6}$$

where $\lambda$ is a balancing factor, $\text{div}(\cdot)$ is given by the diversity between the nodes (Zhou et al., 2017b), defined as $\text{div}(x_i, x_j) = \sum_{c=1}^{M} (P^c(x_i) - P^c(x_j)) \log \frac{P^c(x_i)}{P^c(x_j)}$ with $M = 2$, $P^1(x_i) = \mathbb{E}(x_i)$ and $P^2(x_i) = 1 - P^1(x_i)$ for our binary case. We opt for an additive weighting, instead of a multiplicative one, because the GCN can take advantage of connections with both similar and dissimilar nodes in the learning process, and using a multiplicative weighting could cut dissimilar connections. We found out that the diversity can indirectly bring information about the similarity of two nodes, in terms of class probability.

## 3. Experiments and Results

We validate our method refining the output of a 2D CNN in the tasks of pancreas and spleen segmentation. We compare this approach with the refinement obtained from a conditional random field method (Krähenbühl and Koltun, 2011). Then, we evaluate the effects of different uncertainty thresholds $\tau$ in our refinement method. We also investigate how the number of training examples used to train the base CNN affects our refinement strategy. Finally, we analyze the relationship between the main components used to construct the graph and the refined segmentation obtained. We make our code publicly available for reproducibility purposes[1].

### 3.1. Datasets

We tested our framework using two CT datasets for pancreas, and spleen segmentation. For the pancreas segmentation problem, we used the NIH pancreas dataset[2] (Roth et al., 2016, 2015; Clark et al., 2013). We randomly selected 45 volumes of the NIH dataset for training the CNN model and reserved 20 volumes for evaluating the uncertainty-based GCN refinement. For spleen, we employed the spleen segmentation task of the medical segmentation decathlon (Simpson et al., 2019) (MSD-spleen[3]). For this problem, we trained the CNN on 26 volumes and reserved 9 volumes to test our framework. The MSD-spleen dataset contains more than one foreground label in the segmentation mask. We unified the non-background labels of the MSD-spleen dataset into a single foreground class since we evaluate our method for refining a binary segmentation model.

### 3.2. Implementation Details

#### 3.2.1. CNN BASELINE MODEL

We chose a 2D U-Net to be our CNN model (Ronneberger et al., 2015). We included dropout layers at the end of every convolutional block of the U-Net, as indicated by the MCDO method. The U-Net was trained considering a binary segmentation problem. Since we are employing a 2D model, we trained the models using axial slices. At inference time, we predicted every slice separately and then we stacked all the predictions together to obtain a volumetric segmentation (a similar strategy was used to perform the uncertainty analysis). As a post-processing step, we compute the largest connected component in the prediction, to reduce the number of false positives. At this point, it is worth mentioning that the U-Net was used only for testing purposes and different architectures can be used instead. This is mainly because our refinement method uses the model-independent MCDO analysis.

#### 3.2.2. UNCERTAINTY ANALYSIS AND GCN IMPLEMENTATION DETAILS

We utilized MCDO to compute the expectation and entropy using a dropout rate of 0.3 and a total of $T = 20$ stochastic passes. To obtain volumetric uncertainty from a 2D model, we performed the uncertainty analysis on every individual slice of the input volume and then

---

1. https://github.com/rodsom22/gcn_refinement

2. https://wiki.cancerimagingarchive.net/display/Public/Pancreas-CT

3. http://medicaldecathlon.com/

Table 1: Average dice score performance (%) of the GCN refinement compared with the CNN prediction and a CRF-based refinement of the CNN prediction. Results for pancreas and spleen are presented.

| Task | CNN 2D U-Net | CRF refinement | GCN Refinement (ours) |
|---|---|---|---|
| Pancreas | $76.9 \pm 6.6$ | $77.2 \pm 6.5$ | $\mathbf{77.8 \pm 6.3}$ |
| Spleen | $93.2 \pm 2.5$ | $93.4 \pm 2.6$ | $\mathbf{95.1 \pm 1.3}$ |

we stacked all the results together to obtain the volumetric expectation and entropy. We tested different values for the uncertainty threshold $\tau$ (see section 3.5).

The GCN model is a two-layered network with 32 features maps in the hidden layer and a single output neuron for binary node-classification. The graphical network is trained for 200 epochs with a learning rate of $1e-2$, binary entropy loss, and the Adam optimizer. We kept these same settings for the refinement of both segmentation tasks. After the refinement process, we can replace only the uncertain voxels with the GCN prediction, or we can replace the entire CNN prediction with the GCN output. We use the second approach since we found it producing better results.

### 3.3. Comparison with State of the Art and Baseline CNN

We applied our refinement method independently on every individual sample from the 20 NIH and 9 MSD-spleen testing volumes. Since CRF is a common refinement strategy, we use the publicly available implementation of the method presented in (Krähenbühl and Koltun, 2011) to refine the CNN prediction. This CRF method assumes dense connectivity. Similar to (Krähenbühl and Koltun, 2011), we set one unary and two pairwise potentials. We use the prediction of the CNN as the unary potential. The first pairwise potential is composed of the position of the voxel in the 3D volume. The second pairwise potential is a combination of intensity and position of the voxels. For the CRF refinement, we considered the same ROI used by the GCN.

Results are presented in Table 1. The GCN-based refinement outperforms the base CNN model and the CRF refinement by around 1% and 0.6% respectively in the pancreas segmentation task. For spleen segmentation, our GCN refinement presented an increase in the dice score of 2% with respect to the base CNN, and 1.7% with respect to the CRF refinement. Figs. 2 and 3 show visual examples of the GCN refinement compared with the base CNN prediction.

### 3.4. Influence of the Number of Training Samples

We also evaluate the performance of the GCN refinement when the base CNN is trained with a small number of samples. For this, we randomly selected 10 out of the 45 training samples of the NIH dataset. For spleen, we selected nine. Results are presented in Table 2. Note the increment in the standard deviation of all the models. A reason for this can be that the CNN does not generalize adequately to the testing set, due to the small number of

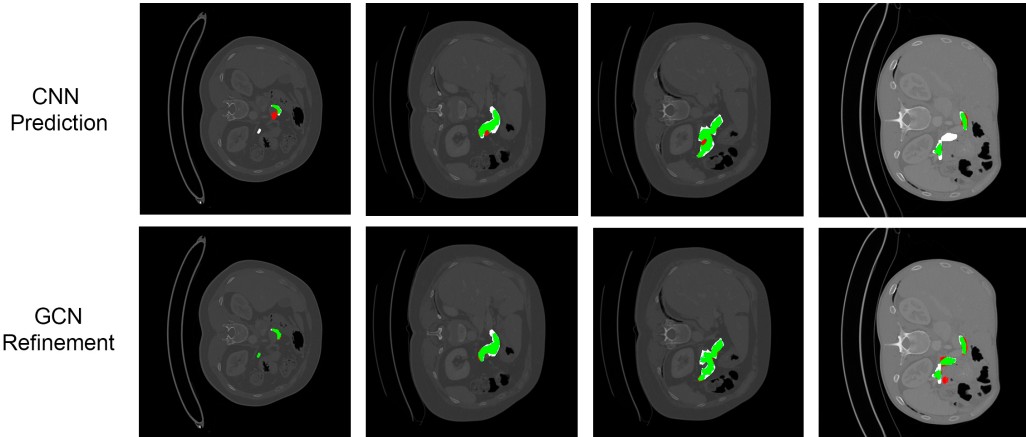

Figure 2: Comparison of the CNN prediction and its corresponding GCN refinement for pancreas segmentation. Green colors indicate true positives (TP), red indicates false positives (FP), and white false negative (FN) regions. From left to right: the first column shows an FP region removed and an FN region recovered after the refinement. The second and third columns show FP regions removed. The fourth column shows an FN region recovered but also a new FP region generated.

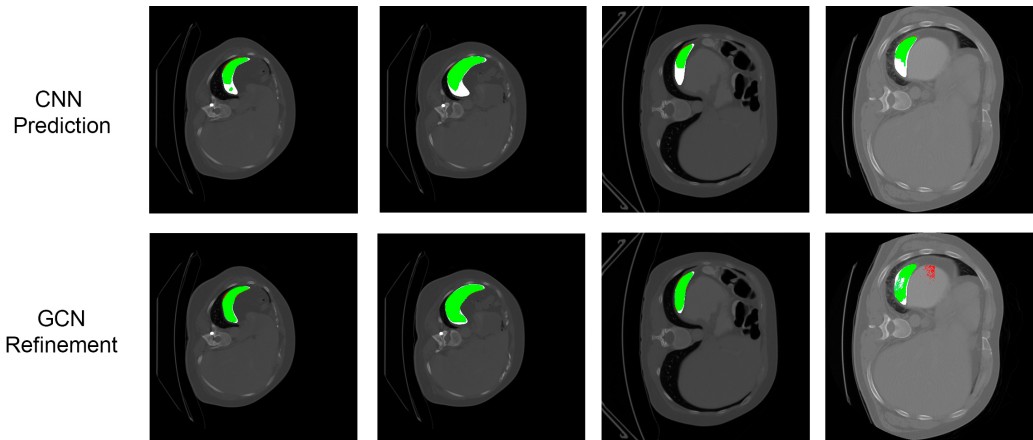

Figure 3: Comparison of the CNN prediction and its corresponding GCN refinement for spleen segmentation. Green colors indicate true positives (TP), red indicates false positives (FP), and white false negative (FN) regions. From left to right: the first, second and third columns show FN regions recovered. The fourth column shows an FN region recovered but also a new FP region generated.

Table 2: Average dice score performance (%) of the GCN refinement compared with the CNN prediction. The CNN model was trained with 10 samples for the pancreas and 9 for the spleen.

| Task | CNN 2D U-Net | CRF refinement | GCN Refinement (ours) |
|---|---|---|---|
| Pancreas-10 | $52.10 \pm 22.61$ | $52.20 \pm 22.62$ | $\mathbf{54.50 \pm 22.15}$ |
| Spleen-9 | $78.80 \pm 28.40$ | $78.80 \pm 28.40$ | $\mathbf{81.15 \pm 28.90}$ |

Table 3: Average dice score performance (%) of the GCN refinement at different uncertainty thresholds $\tau$. Pancreas-10 and Spleen-9 indicate the models trained with 10 and nine samples, respectively.

| Task | GCN $\tau = 1e-3$ | GCN $\tau = 0.3$ | GCN $\tau = 0.5$ | GCN $\tau = 0.8$ | GCN $\tau = 0.999$ |
|---|---|---|---|---|---|
| Pancreas | $77.71 \pm 6.3$ | $77.79 \pm 6.4$ | $77.77 \pm 6.3$ | $77.81 \pm 6.3$ | $77.79 \pm 6.3$ |
| Pancreas-10 | $54.55 \pm 22.1$ | $54.32 \pm 22.1$ | $54.15 \pm 22.2$ | $53.91 \pm 22.4$ | $53.14 \pm 22.9$ |
| Spleen | $95.01 \pm 1.5$ | $94.92 \pm 1.4$ | $94.98 \pm 1.4$ | $94.97 \pm 1.4$ | $95.07 \pm 1.3$ |
| Spleen-9 | $80.91 \pm 28.8$ | $80.94 \pm 28.9$ | $80.94 \pm 28.8$ | $80.98 \pm 28.9$ | $81.15 \pm 28.9$ |

training examples. Similar to the previous results, the increment in the dice score for the GCN refinement is about 2.4% with respect to the CNN base model for the pancreas, and improvement of 2.3% for spleen, compared with the base CNN.

### 3.5. Influence of Uncertainty Threshold

In our experiments, we evaluate the influence of different values for $\tau$. We tested the method with values of $\tau \in \{0.001, 0.3, 0.5, 0.8, 0.999\}$. In this way, we covered a wide range of conditions that define a voxel as "uncertain". After training the GCN, we replaced all the CNN predictions with the GCN output. Table 3 compares the CNN output with the GCN refinement at different values of $\tau$ for the tasks of the pancreas and spleen segmentation.

The parameter $\tau$ controls the minimum requirement to consider a voxel as uncertain. Lower values lead to a higher number of uncertain elements. This has a direct relationship with the number of high certainty nodes in the graph representation, and hence, in the number of training examples for the GCN. This also influences the quality of the training voxels for the GCN, since a high threshold relaxes the amount of uncertainty necessary to rely on the prediction of the CNN.

However, from the results of Table 3, except for pancreas-10 and spleen-9, there is no significant impact on the choice of this parameter. One reason can be that there is a clear separation between high and low uncertainty points. Therefore, changing $\tau$ may add (remove) a few number of nodes that are insignificant for the learning process of the GCN.

For the pancreas-10 model, we notice a progressive decrease in the dice score. Since this model uses fewer training examples, it is expected to have low confidence in their predictions

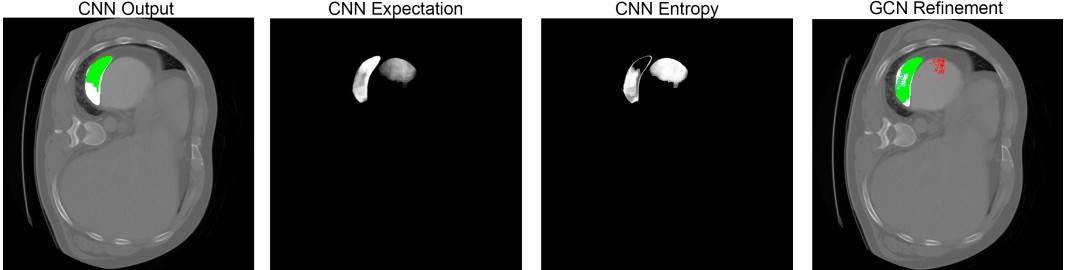

CNN Output     CNN Expectation     CNN Entropy     GCN Refinement

Figure 4: Elements used in the graph definition. In the CNN and GCN outputs: Green colors indicate true positives, red false positives, and white false negative regions. For the expectation and entropy: brighter intensities indicate higher values.

(in contrast with the model trained with 45 volumes). In this scenario, a higher uncertainty threshold increases the chance to include high-uncertainty nodes as ground truth for training the GCN. A lower $\tau$ includes fewer points but with higher confidence. This appears to be beneficial in the pancreas segmentation model trained with fewer examples.

The opposite occurs with spleen-9, where higher $\tau$ are beneficial. This might indicate a dependency on the characteristics of the anatomies since the pancreas presents more inter-patient variability.

In general, our results suggest that $\tau$ parameter should be selected based on the target anatomy. Further, $\tau$ appears to have more influence in conditions of high uncertainty, e.g. when the model is trained with fewer examples. In the cases where $\tau$ has no significant impact, intermediate values are preferred, since they lead to a lower number of nodes, and in consequence to lower memory requirements.

### 3.6. Deep Insights on Prediction, Expectation, and Entropy

We employed three elements from the uncertainty analysis in the definition of our graph: the CNN's prediction, the CNN's expectation, and the CNN's entropy. Fig. 4 shows an example of these components.

The labels of the graph are given by the CNN's high-confidence prediction. However, from Fig. 4 we can see that the refinement is similar to the expectation. The expectation is one of the features of the nodes. Also is the main component for the diversity in the edge's weighting function (see section 2.2.2). The GCN can learn how to use the CNN's expectation, together with intensity and spatial information, to reclassify the nodes of the graph. However, it can also generate false positives if the expectation contains artifacts. Fig. 4 shows an example of this case, where we can see a region in the expectation that does not agree with the ground truth. It can be also noticed that the GCN reduced this region. This can be a result of the random long-range connections included in the graph definition.

In our last experiment, we evaluate the relationship between the expectation and the GCN refinement. For this, we compute the relative improvement between the GCN and

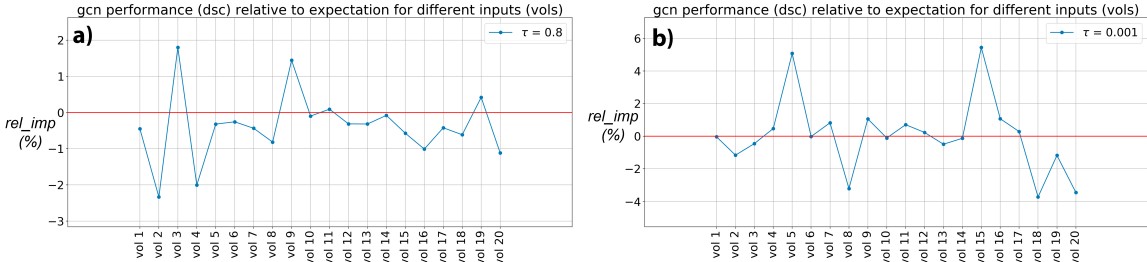

Figure 5: Relative improvement (%) per input volume of different GCN configurations respect to the expectation of a pancreas segmentation model. The red line indicates the same dsc as the expectation. a) CNN trained with 45 volumes, $\tau = 0.8$. b) CNN trained with 10 volumes, $\tau = 0.001$.

the expectation. First, the expectation was thresholded by 0.5. Then we computed its dice score with the ground truth. The relative improvement is computed as:

$$rel\_imp = \frac{gcn_{dsc} - expectation_{dsc}}{expectation_{dsc}} \times 100. \qquad (7)$$

We compute $rel\_imp$ for every input volume. Fig. 5 shows the results for the pancreas segmentation task, and compares the metric when the expectation was obtained from a model trained with 45 (Fig.5a) and 10 samples (Fig.5b), respectively, for pancreas segmentation.

Fig. 5a shows that most of DICE coefficients (17/20) of the GCN refinement are either below or close to the ones of the expectation. However, three volumes show an improvement in the DICE compared to the expectation. This is different in Fig. 5b. Here, (13/20) volumes show either better or similar DICE for the GCN compared to the expectation. A possible explanation is that models trained with adequate number of examples (volumes), their expectation is good enough. In contrast, models trained with a few examples (volumes) have higher uncertainties yielding unreliable expectations. Our results suggest that our GCN refinement strategy is favourable over the expectation or uncertainty analysis in such scenarios.

## 4. Discussion and Conclusion

In this work, we have presented a method to construct a sparse semi-labeled graph representation of volumetric medical data, based on the output and uncertainty analysis of a CNN model. We have also shown that graph semi-supervised learning can be used to obtain a refined segmentation. Future research can be directed in the following directions:

**2D vs. 3D Models:** In our experiments, we employed a 2D architecture since it provides us with all necessary components to test out method. Nevertheless, our refinement strategy is orthogonal to other segmentation approaches and can be applied to any CNN that produces uncertainty measures with MCDO. This also includes 3D models. However, to our best knowledge, MCDO is most commonly employed with 2D models and its translation to

3D might require additional methodological efforts derived from working with 3D architectures together with additional requirements for data-handling due to memory constraints (LaBonte et al., 2019). Further, 3D model might not necessarily provide a better initial segmentation compared to a 2D CNN as reported by (Zhou et al., 2019; Wang et al., 2019). Nevertheless, investigating our approach using 3D models equipped with MCDO might be an interesting direction.

**Uncertainty Quantification:** In this work, we have employed MCDO (Kendall and Gal, 2017) for the model uncertainty analysis, and found out the expectation could be a good choice for well-trained models, while our GCN refinement show superior performance, compared to the expectation, in low-data regime. Nonetheless, recent proposed uncertainty measures (Tomczack et al., 2019), which disentangle the models uncertainty from the one associated with the inter/intra-observer variability, might be desirable.

**Graph Representation:** We have investigated different connectivity and weighting mechanisms in defining our graph, and extracted a couple of features to represent our nodes. However, prior knowledge, e.g. geometry, could be used to constrain the ROI and provide plausible configurations (Degel et al., 2018; Oktay et al., 2017).

## Acknowledgments

R. D. S. is supported by Consejo Nacional de Ciencia y Tecnología (CONACYT), Mexico. S.A. is supported by the PRIME programme of the German Academic Exchange Service (DAAD) with funds from the German Federal Ministry of Education and Research (BMBF).

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
