# OpenReview forum: "Uncertainty-based Graph Convolutional Networks for Organ Segmentation Refinement"
_MIDL.io/2020/Conference — MIDL 2020_

### Official Review · AnonReviewer1 · 2020-03-09
**A GCN based segmentation refinement method considering uncertain regions from CNN.**

**Rating:** 4
**Confidence:** 4
**Recommendation:** Oral

**Summary:**

In this paper, the authors propose a two-step segmentation refinement algorithm. In the first step, an uncertainty analysis is performed on the predictions from a CNN network. The Monte Carlo dropout technique is applied to CNN prediction to obtain the uncertain regions. Next, a semi-labelled graph is built based on intensity, entropy, uncertainty and the output of CNN. This constructed graph is then used to train a GCN and further refine the segmentation.

**Strengths:**

1. The authors validate the proposed method on two CT datasets segmentation pancreas and spleen.
2. The paper compares the proposed work against a 2D CNN (UNet) and a CRF based refinement method.
3. The paper also presents the results describing the effect of training samples on the trained CNN.
4. Extensively evaluate the refinement method for different threshold values.
5. Evaluate the effect of the graph construction process with the final segmentation obtained.

**Weaknesses:**

Minor
1. What is the computation gain over the CRF based refinement?
2. Is the training end to end? If not, what is the stopping criteria for the CNN? Since GCN seems to refine the segmentation better, did authors consider early stopping the CNN and train a larger GCN model for refinement?
3. Increase the legends of the figures (1 and 5) and text size of the axis. Hard to read.

**Justification Of Rating:**

The paper is well written. The authors justify the claims made in the paper. The proposed method is evaluated on two datasets. The effect of the different threshold values is studied. The results are compared with a dense CRF method.

**Paper Type:**

methodological development

**Special Issue:**

yes

---

> ### Author Response · Authors · 2020-03-27
> **Answer**
>
> We would like to thank the reviewer for his constructive feedback. Next, we address the reviewer’s minor questions:
>
> 1. Our method requires to define a graph and then train a GCN model. This can require more computational requirements compared with the most efficient versions of CRF. In our experiments, the time required for training and testing the GCN in the constructed graph is around twice the time required for CRF in the fully connected version of our uncertainty graph. This is a time of around 30 sec ~1 min for the GCN vs. 13 sec ~ 30 sec for CRF. These numbers do not consider the time for uncertainty analysis and graph construction.
>
> 2. Our model is a refinement strategy that works on the predictions, and uncertainty estimates of any off-the-shelf ConvNets. This means our model requires a CNN that was previously trained using MCDO for uncertainty estimations. Hence, our GCN model is not trained in end-to-end fashion (jointly with the CNN). Since our aim is not to replace current employed models, however, complement them, we are assuming that the CNN is trained in a standard procedure. In practice, we train the CNN until convergence and keep the best performing model on the validation set. Applying an early stopping criterion could have as a consequence an increase in the uncertainty area, generating a bigger ROI, increasing the number of nodes and, hence, tremendously increasing the memory requirements for the GCN. The required resources can vary in function of the selected number of epochs and can behave differently for different organs. For example, an early stopping at the epoch 10 generates an increase of 20 % in time processing (due to an increasing number of nodes) on the NIH pancreas, while the same number of epochs makes the graph not able to fit in GPU memory for the decathlon spleen dataset.
>
> 3. Thanks for spotting this. We have just updated the figures, accordingly.

---

### Official Review · AnonReviewer2 · 2020-03-13
**Interesting work as post-processing step for segmentation refinement**

**Rating:** 3
**Confidence:** 4
**Recommendation:** Poster

**Summary:**

The authors proposed a GCN-based post-processing step for segmentation refinement.  Segmentation network can provide useful information about potentially mis-classified elements and the later GCN can be trained in a semi-supervised way to refine the segmentation. Compared with CRF, the GCN refinement can have 0.6% and 1.7% Dice improvements on Pancreas and Spleen dataset, respectively.  More improvements are reported when use less samples for training and thus CNN does not generalized adequately to the unseen testing data.

**Strengths:**

1. The idea of GCN refinement strategy is novel and interesting.
2. The authors provided detailed ablation study on GCN refinement by reducing training sample size, changing threshold
3. The authors gave comprehensive discussion about deep insights behind the proposed model which could help readers understand their model well.

**Weaknesses:**

1. It is not clear how its flexibility to work with other advanced segmentation models. The authors used 2D U-Net as an example, is your model available to work with 3D models ?
2. The focus of this paper is about this GCN refinement method which may limit its use in reality. I think recent segmentation models can have better results than the authors reported using 2D U-NET + the proposed refinement. It is better to compare with more recent segmentation models.

**Detailed Comments:**

1. GCN is used as post-processing way for refinement. there are other works to use shape prior jointly with segmentation to make the model have ability to incorporate with organ geometry information. It is recommended to include their work in your literature review or compare them in your experiment section.

For example,
Prior-aware Neural Network for Partially-Supervised Multi-Organ Segmentation, ICCV 2019
Integrating 3D Geometry of Organ for Improving Medical Image Segmentation, MICCAI 2019

**Justification Of Rating:**

1. The use of GCN refinement is novel even though it is not clear how this can work with more advanced segmentation models.
2. It is very helpful in the task of using 2D models for segmentation. The authors prove it can have better refinement than widely used CRF.

**Paper Type:**

validation/application paper

**Questions To Address In The Rebuttal:**

1. If the proposed refinement strategy can work with other more advanced models, like 3D models.

**Special Issue:**

no

---

> ### Author Response · Authors · 2020-03-27
> **Answer**
>
> We would like to thank the reviewer of the constructive feedback:
>
> 1. We addressed the topic in the general comments.
>
> 2. Thanks for the suggestions. We have added the suggested literature to our related work. As reported in (1), our method is orthogonal to other segmentation techniques, and we don’t claim a state-of-the-art performance here. We do claim the superior performance of our refinement step compared to SOTA methods, i.e. Conditional Random Field (CRF). Indeed, recent segmentation methods that incorporate prior knowledge, i.e. shape prior, are loosely relevant to our work.

---

### Official Review · AnonReviewer4 · 2020-03-13
**GCN based refinement of organ segmentation which utilizes voxel level uncertainty measures, with comprehensive experiments.**

**Rating:** 3
**Confidence:** 5
**Recommendation:** Poster

**Summary:**

Graph convolution network (GCN) based refinement of organ segmentations in 3D is proposed. Uncertainty information is derived using Monte Carlo drop-out on the U-Net predictions and a graph using the uncertainty and entropy information is constructed. Some of the nodes in the uncertainty graph are then labeled to indicate their uncertainty level. Uncertainty levels of the unlabeled nodes are inferred using a standard GCN. Performance of the method is compared with baseline U-Net and the refinement using uncertainty is compared with a CRF based method. Experiments are conducted on two datasets for 2d segmentation.

**Strengths:**

+ The idea of constructing uncertainty graphs and partially labeling them is an interesting contribution
+ Use of GCNs in this setting is novel
+ Relevant comparision with the fully connected CRFs for the uncertainty refinement strategy
+ Experiments on two datasets show reasonable improvements
+ Thorough discussions about the influence of different parameters (tau, number of samples) are presented


**Weaknesses:**

1. The method is motivated for 3D but both experiments are presented using 2D Unet. Why was this choice made? Using a 3D segmentation model would be more natural in this setting.

2. The selection of node neighbourhoods by adding additional 16 neighbours beyond the nearest neighbours is interesting. However, what is the motivation to do this? How do these 16 random neighbours contribute to the GCN? If no attention was used when performing the GCN updates these additional neighbours might hamper the learning.

3. Are the results in Table 1 significant?

4. The discussion in Sec 3.4 and Table 2 is interesting but as the authors point out the standard deviation is large. Do your conclusions hold about the improvements due to GCN? Again, are these bold face numbers significant improvements?

**Detailed Comments:**

See Strengths and Weaknesses above.

**Justification Of Rating:**

The idea of using GCNs to refine segmentations based on uncertainty is certainly interesting. However, a couple of key ideas (why 2d segmentation model, choice of random neighbours, significance of results) need to be further clarified.

**Paper Type:**

methodological development

**Questions To Address In The Rebuttal:**

Addressing Points 1, 2, 3 in Weaknesses listed above are essential to be addressed.

**Special Issue:**

no

---

> ### Author Response · Authors · 2020-03-27
> **Answer**
>
> We would like to thank the reviewer of the constructive feedback:
>
> 1. We have addressed this topic in the general comments.
>
> 2. We have addressed this topic in the general comments.
>
> 3. It is worth mentioning that our testing set is quite small (9 for spleen, and 20 for pancreas), and oftentimes such statistical significance tests simply fail (see [1, 2]). Nevertheless, to increase the sample size, we generate the dice score in slice-wise, so we end up with 278-1700 slices, then we have run the non-parametric statistical significance test, namely  Kolmogorov–Smirnov test for all our experiments reported in Table 1 and 2. Our results are statistically significant in all results except the one of the spleen dataset in Table 1.
>
>
>                  |             Table 1 (full)                    |       Table 2 (limited)
> ------------------------------------------------------------------------------------------------------
>                  |     GCN vs CNN  |  GCN vs CRF |   GCN vs CNN  |  GCN vs CRF
> ------------------------------------------------------------------------------------------------------
> Spleen     |        Not              |  Not                | KS (p < 0.001)  |    KS (p < 0.001)
> ------------------------------------------------------------------------------------------------------
> Pancreas |     KS (p < 0.06) | KS (p < 0.06)  | KS (p < 0.05)    |    KS (p < 0.05)
> ------------------------------------------------------------------------------------------------------
>
>
> [1] Biau DJ, Kernéis S, Porcher R. Statistics in brief: the importance of sample size in the planning and interpretation of medical research. Clin Orthop Relat Res. 2008;466(9):2282–2288. doi:10.1007/s11999-008-0346-9
>
> [2] Szucs D, Ioannidis JPA. When Null Hypothesis Significance Testing Is Unsuitable for Research: A Reassessment. Front Hum Neurosci. 2017;11:390. Published 2017 Aug 3. doi:10.3389/fnhum.2017.00390

---

> > ### Comment · AnonReviewer4 · 2020-04-01
> > **Satisfactory response to reviews; Increasing score to "Strong Accept"**
> >
> > Major concerns (points 1 & 2) have been satisfactorily addressed.
> >
> > Regarding the test set size: I agree the test set is too small. I missed that detail earlier. I would urge the authors to include the details about using 2d slices extracted from 3d volumes for clarity. As far as I see, this detail is missing in the submitted version.
> >
> > Based on the authors' rebuttal, I am raising the score to "Strong Accept."

---

### Official Review · AnonReviewer3 · 2020-03-20
**The author presented an uncertainty-based graph convolutional neural networks for organ refinement of the CNN based segmentation. It’s an innovative methods and combination to improve the traditional CNN based segmentation.**

**Rating:** 4
**Confidence:** 4
**Recommendation:** Best Paper Award, Oral

**Summary:**

The author innovatively combines the CNN and GCN together to improve the segmentation results of the CNN. According to the author, refinement strategy is not only limited to specific underlying segmentation model but also generalizable. The segmentation is always a challenging problem needs continuous improvement. The authors provided enough experiment results with multiple use cases. Some minor details might be provided to address further concerns from readers

**Strengths:**

The combination of proposed approaches in the paper is innovative
The paper is well written and background introduction and discussion is adequate
The experiment design is well considered with comparison.

**Weaknesses:**

The abstract is less informative, not sure if it’s due to word limits. But could be better organized to reflect the imaging modality, dataset and results detail. Also, there is no definition of abbreviation.
Introduction part, the abbreviation is defined multiple time, for example CNN
For formula 3 the indicator definition is confusing, is it just a thresholding function. Not sure why use square bracket rather than
If the author could provide more detail about connection establishment of the between nodes that would be helpful. For example, why perpendicular immediate neighbors need to be connected. What if the voxel is on boundary so the connected two nodes might fall into different labels. Also, why choose randomly 16 nodes in the graph. Does the distance between node and node original intensity will affect the weight of the connection?


**Justification Of Rating:**

The overall paper is well written and detailed. The approaches proposed in the paper is innovative in its specific domain and use case.  Some details about the graph definition, parameter choosing and abbreviation definition are minor issues could be addressed.

**Paper Type:**

methodological development

**Special Issue:**

no

---

> ### Author Response · Authors · 2020-03-27
> **Answers**
>
> First of all, we would like to thank the reviewer for the constructive feedback.
>
> 1- A brief motivation and clarification have been already added to the revised manuscript. Further, we have updated our abstract and make it more informative.
> 2- Regarding connectivity, it has been addressed in the general comments
> 3- Indeed, our weighting function is carefully designed to highly consider the distance between pixels in their space and intensity when they share similar probabilities, e.g.ambiguous cases. For example, when we have pixels with high uncertainty, this yields to predictions around 0.5, and almost zero diversity. In this case, the distance in intensity and space will dominate the weight.

---

### Author Response · Authors · 2020-03-27
**General Comments**

We would like to thank all the reviewers for their efforts to provide constructive feedback to our work. According to the received comments, we present the following general insights about our work.

Regarding the applicability to different models, including 3D ConvNets

Our refinement strategy is orthogonal to other segmentation approaches and can be applied to any ConvNets that produce uncertainty measures with MCDO. In principle, this should also include 3D models compatible with MCDO. However, to our best knowledge, MCDO is most commonly employed with 2D models and its translation to 3D ConvNets might require additional methodological efforts derived from working with 3D architectures. Further, 3D ConvNet usually comes with additional requirements for data-handle due to memory constraints. For example, the subdivision of the input volume in overlapping chunks [1]. Additionally, in terms of performance, a 3D model will not necessarily provide a better initial segmentation compared with a 2D ConvNet (see [2 ,3,4]), as pointed by [2]:  "[...] 2D models generally observe a better performance in each setting compared with 3D models. This is probably due to the fact that current 3D models only act on local patches (e.g., 64 × 64 × 64), which results in lacking holistic information [...]".

Regarding the Connectivity

The most straightforward option is to consider the connectivity with adjacent voxels (6 or 26 adjacent voxels). However, this simple nearest neighborhood scheme may not be adequate in our problem for two reasons; First, with this scheme, every single voxel is connected with its local neighborhood but lacks global information. Second, as presented in Fig. 4 in our paper, voxels with high uncertainty tend to shape contiguous clusters. With a simple nearest neighborhood scheme, voxels inside these clusters will be only connected to their adjacent neighbors, i.e. uncertain voxels, with almost no connection with voxels with high confidence. Voxels living in the boundary of these clusters are the only ones who are connected to voxels with high confidence. Hence, the propagation of information from voxels with high confidence to the uncertain voxels will be somehow limited. To overcome this limitation, one could think of dense connections where each voxel is connected to all voxels, however, it is computationally expensive. Therefore, we opt to include 16 random long-distance connections between the nodes, and we show the effectiveness of this approach.

References
[1] Tyler LaBonte, Carianne Martinez, and Scott A. Roberts. We Know Where We Don't Know: 3D Bayesian CNNs for Uncertainty Quantification of Binary Segmentations for Material Simulations. arXiv:1910.10793v1
[2]  Prior-aware Neural Network for Partially-Supervised Multi-Organ Segmentation, ICCV 2019
[3] Matthew Lai. Deep Learning for Medical Image Segmentation. arXiv:1505.02000v1
[4] Yan Wang, et al., Abdominal Multi-organ Segmentation with Organ-Attention Networks and Statistical Fusion. arXiv:1804.08414v1

---

### Meta-Review · Area_Chair1 · 2020-04-06
**MetaReview of Paper146 by AreaChair1**

**Rating:** 4
**Recommendation For Accepted Papers:** Oral

**Metareview:**

All the reviewers recommended acceptance of this work. After reading their comments and the answer given by the authors in the rebuttal, I think this work can be accepted for publication at MIDL.

Please, when submitting the Camera Ready version, take into account the suggestions made by the reviewers.

**Paper Type:**

methodological development

**Special Issue:**

no

---

### Decision · Program_Chairs · 2020-04-11

Accept